# On the Mechanism and Kinetics of Synthesizing Polymer Nanogels by Ionizing Radiation-Induced Intramolecular Crosslinking of Macromolecules

**DOI:** 10.3390/pharmaceutics13111765

**Published:** 2021-10-22

**Authors:** Aiysha Ashfaq, Jung-Chul An, Piotr Ulański, Mohamad Al-Sheikhly

**Affiliations:** 1Department of Chemistry and Biochemistry, University of Maryland, College Park, MD 20742, USA; aiysha@umd.edu; 2Anode Materials Research Group, Research Institute of Industrial Science & Technology (RIST), Pohang 37673, Korea; jcan@rist.re.kr; 3Institute of Applied Radiation Chemistry, Faculty of Chemistry, Lodz University of Technology, Wroblewskiego 15, 93-590 Lodz, Poland; 4Department of Materials Science and Engineering, University of Maryland, College Park, MD 20742, USA

**Keywords:** nanogels, ionizing radiation, intramolecular crosslinking, radiation induced synthesis nanogels

## Abstract

Nanogels—internally crosslinked macromolecules—have a growing palette of potential applications, including as drug, gene or radioisotope nanocarriers and as in vivo signaling molecules in modern diagnostics and therapy. This has triggered considerable interest in developing new methods for their synthesis. The procedure based on intramolecular crosslinking of polymer radicals generated by pulses of ionizing radiation has many advantages. The substrates needed are usually simple biocompatible polymers and water. This eliminates the use of monomers, chemical crosslinking agents, initiators, surfactants, etc., thus limiting potential problems with the biocompatibility of products. This review summarizes the basics of this method, providing background information on relevant aspects of polymer solution thermodynamics, radiolysis of aqueous solutions, generation and reactions of polymer radicals, and the non-trivial kinetics and mechanism of crosslinking, focusing on the main factors influencing the outcomes of the radiation synthesis of nanogels: molecular weight of the starting polymer, its concentration, irradiation mode, absorbed dose of ionizing radiation and temperature. The most important techniques used to perform the synthesis, to study the kinetics and mechanism of the involved reactions, and to assess the physicochemical properties of the formed nanogels are presented. Two select important cases, the synthesis of nanogels based on polyvinylpyrrolidone (PVP) and/or poly(acrylic acid) (PAA), are discussed in more detail. Examples of recent application studies on radiation-synthesized PVP and PAA nanogels in transporting drugs across the blood–brain barrier and as targeted radioisotope carriers in nanoradiotherapy are briefly described.

## 1. Introduction

Nanogels (NGs) consist of internally crosslinked macromolecules with solvent filling the spaces between the polymer segments. In size, NGs can measure up to 100 nm (larger particles of this kind are called microgels). NGs have gathered significant interest over the past decade for their applications as nanocarriers for drug delivery, tissue engineering, biomedical implants, and gene therapy. Due to their good biocompatibility, high stability, hydrophilic and drug loading properties, nanogels can make ideal nanocarriers. The hydrophilic nature of some NGs is due to the presence of polar groups along their polymer chains (-OH, -COOH, -CONH_2_, etc.). The properties of nanogels can be controlled by varying the type and concentration of polymer used in synthesis [1]. Drugs can be loaded and released at target sites through internal and external stimuli. Nanogels also have the ability to encapsulate biological compounds like proteins, DNA and RNA inside their polymer networks. In aqueous media, the polymers swell to an equilibrium but retain their shape despite the large amounts of liquid within their structures. This unique property of swelling but maintaining their shape, along with their structural flexibility and elasticity, are of great interest to investigators. They are often referred to as “frozen” polymer coils. They can be thought of as macromolecular cages that bind and transport many different molecules within their structure and/or at their surface.

There are several methods that can be employed for the preparation of nanogels. One of the most typical approaches is to run crosslinking polymerization of monomers and crosslinking agents in an inverse microemulsion, where small droplets of water containing the reactants are contained within micelles stabilized by a surfactant [2,3,4]. The reaction may be initiated thermally in the presence of a suitable initiator, by light [5,6] or by ultrasound [7,8,9,10,11]. Alternatively, crosslinking polymerization can be performed in solutions. This requires careful choice of solvent and reaction conditions in order to avoid macrogelation [12,13,14]. Easier process control in such cases can be provided by employing controlled radical polymerization, where initiation can be provoked by irradiation [15,16]. Another synthetic approach involves starting with polymer chains as substrates and inducing intramolecular crosslinking using a chemical crosslinking agent; for instance chains of poly(vinyl alcohol) can be internally crosslinked by glutaraldehyde if proper conditions are chosen [17]. A broader description of classical nanogel and microgel synthesis methods can be found in reviews [13,18,19,20,21]. However, despite this diversity of procedures, many nanogels cannot be used in the field of medicine due to their toxicity. This toxicity arises from potentially hazardous unreacted monomers, surfactants, oligomers and initiators used in many preparation procedures. Irradiation-based methods are gaining a lot of interest as a way to synthesize non-toxic nanogels. This method is based on ionizing radiation and, in some cases, on physical self-assembly techniques which do not require polymerization initiators or crosslinking agents. Essentially the only substrates needed are polymer and water, which greatly eliminates toxicity concerns. Another noteworthy advantage of this method of synthesis is that sterilization occurs simultaneously with the synthesis of the NG.

This review aims to provide readers with rudimentary information on the existing theories and practices of NG formation via ionizing radiation. For this reason, the reviewer will cover a host of topics including a historical summary of water radiolysis, a synthetic overview of NG formation by radiation, the decay kinetics of the formed radicals, and a range of product analysis techniques. Specifically, the review will focus on the radiation-induced synthesis of polyvinylpyrrolidone [22], poly(acrylic acid) [23], and of interpolymer complexes of poly(acrylic acid) with polyvinylpyrrolidone [24] nanogels. Their applications for tumor targeting [25] will be discussed, along with their use as insulin nanocarriers for a new type of treatment for Alzheimer’s disease [26]. Though a goal of this review is to be thorough, it is also the intent of the authors to present the information in a concise and clear manner so that experts and novices alike may benefit. 

## 2. Ionizing Radiation

Nanogels can be synthesized through intramolecular recombination of radicals that are created along their polymer chain from exposure to ionizing radiation. The possibility of this alternative route was first discovered in the 1950s [27]; however, extensive research on radiation synthesis of NGs and their applications as drug delivery agents did not pick up until the 1980s. 

Ionizing radiation is any type of radiation that has enough energy to ionize atoms and molecules. The radiation may come in the form of photons like gamma rays from a Cobalt-60 source or fast-moving particles like the electron beams generated by accelerators. The dose or the amount of energy absorbed by matter is measured in grays, or Gy. The G-value or radiation chemical yield reports the yield of ions, radicals, and stable molecules produced or consumed in units of moles per absorbed energy (mol/J). 

The synthesis of NGs via ionizing radiation in practice simply requires the polymer of interest and the solvent which in the majority of cases is water. Though polymers can also be irradiated in the solid state (i.e., “in bulk”), this review will focus on polymers in aqueous solutions. This is of great significance for biomedical applications since toxic agents or additives are not required. 

The ionizing radiation, whether it be γ-rays or fast electrons, does not directly interact with the polymer in dilute solutions. Rather, most of the radiation energy is absorbed by the water. The water molecules then form a variety of short-lived reactive species that react with the polymeric material. Thus, in order to study the mechanism through which NGs are formed it is vital to understand the radiolysis of water. 

Before proceeding, it is important to note that if a polymer is present in a considerable amount, radiation energy can be deposited directly on the polymer. There are examples of these studies found in reviews [28] and the references therein, but the majority of studies and applications involve dilute or semi-dilute aqueous solutions.

## 3. Historical Overview of Water Radiolysis

Interest in the interaction of radiation and water started almost immediately in the early days of radiation chemistry. It was not long after the discovery of radium in 1898 that Giesel, Skłodowska-Curie and Debierne first reported results that water could decompose under the effects of radium emanation [29,30]. Before this, scientists had already made observations and measurements of the generation of H_2_ and O_2_ gases by aqueous solutions of radium bromide. However, many still did not believe pure water could decompose by irradiation. In the 1930s, Fricke reported the different effects low-energy radiation such as UV and high energy radiation such as X-rays had on aqueous systems. In the case of UV, photons directly interacted with the molecules, while X-rays seemed to “activate” water molecules, which would then react with the solutes present. This concept of “activated water” would be the starting point of our modern understanding of water radiolysis. In 1944, J. Weiss first proposed that water split into H^•^ and ^•^OH radicals which then could interact with themselves and solutes in solutions [31]. The H^•^ radicals were deemed responsible for the observed reductions in solutions and the ^•^OH responsible for the oxidation. In the presence of oxygen, hydrogen peroxide was also formed. The molecular product H_2_ was also observed. It was understood that these species were not formed homogenously throughout the water. Rather, they formed along the particle track of the radiation and then diffused into the rest of the solution [32,33]. It was initially assumed that ejected electrons could not escape the coulombic field of their parent ion, and hence not travel far in solutions. Platzman, Day, Stein and Weiss all suggested that if the electron is thermalized or slowed down, it can become hydrated in aqueous solutions [34,35,36]. In 1962, Hart, Boag and Keene detected the absorption spectrum of the hydrated electron around 700 nm using pulse radiolysis [37,38,39]. A more in-depth dive into the history can be found elsewhere [30,40,41].

The development of reactors during the 1950s expedited developments in radiation chemistry by introducing new and better radiation sources. It was now possible to carry out experiments with higher dose rates and penetrating power. Cobalt-60 was a popular isotopic source that guaranteed pure gamma radiation rather than mixtures of various radiation types, which simplified experimental set-ups. Furthermore, the advent of pulse radiolysis in the 1960s also significantly increased the type of experiments that could be performed and the information that could be obtained. After the discovery of various radical and molecular products, the focus shifted towards studying radiation chemical yields and the rate constants of these reactions [42,43,44,45]. Pulse radiolysis allowed time resolutions to be decreased from the order of microseconds to eventually, now, picoseconds. By 1967, rate constants for over one thousand different reactions induced directly or indirectly by radiation of inorganic and organic compounds in aqueous solutions had been compiled [46] and since then many more new data have been made available [47,48]. 

## 4. Mechanisms of Water Radiolysis

Since there are several types of ionizing radiation (electrons, photons, neutrons, etc.), energy depositions are highly dependent on the type of radiation in question. 

The interaction of radiation with water starts with an energy deposition which initially forms ionized water molecules (H_2_O^+^), excited water molecules (H_2_O*) and sub-excitations electrons (e^−^) [47,49,50]. This initial step is referred to as the *physical stage*, and occurs in about 0–10^−15^ s. The next step, known as the *physico-chemical stage*, involves a whole host of fast reactions (10^−15^–10^−12^ s), where molecules are rearranged and electrons are thermalized and hydrated. The in-depth description of these reactions is beyond the scope of this text. It is in this stage that radicals first appear in our system. The final stage is the *chemical stage* (10^−12^–10^−9^ s), where the species diffuse in solution and can react with solutes present in the system. Figure 1 summarizes the three stages of water radiolysis. 

The general equations for water radiolysis are:H_2_O → e_aq_^−^, ^•^OH, H^•^, H_2_O_2_, H_2_, H_3_O^+^, OH^−^(1)
N_2_O + e_aq_^−^ + H_2_O → ^•^OH + OH^−^ + N_2_(2)
H_3_O^+^ + e_aq_^−^ → H^•^ + H_2_O(3)

Reactions (2) and (3) are diffusion controlled, their rate constants being 9.1 × 10^9^ L mol^−1^ s^−1^ [51] and 2.3 × 10^10^ L mol^−1^ s^−1^ [52], respectively. For nanogel synthesis as described in this work, ^•^OH radicals are of special importance. In neutral and alkaline aqueous solutions, their yield can be doubled by saturating solutions with nitrous oxide (reaction 2) [53]. In acidic media reaction (2) is of less importance due to the competing reaction (3).

Hydroxyl radicals, when generated in aqueous solutions of organic aliphatic compounds and in particular simple water-soluble polymers, react with them mainly by abstracting hydrogen atoms. As a result, carbon-centered polymer radicals are formed, and water molecules are re-created (reaction 4). For dilute polymer solutions, where the direct interaction of radiation with macromolecules can be neglected, this indirect process via ^•^OH radicals is the main source of polymer radicals. H-atoms can also abstract hydrogen from macromolecules, but in typical conditions this reaction is of lesser importance since the radiation-chemical yield of H^•^ is significantly lower than that of ^•^OH (unless irradiation is performed in strongly acidic solutions, see reaction 3), and the rate constants of hydrogen abstraction reactions by H^•^ are lower than those by ^•^OH.
R-H + ^•^OH (H^•^) → R^•^ + H_2_O (H_2_)(4)

Reactions of carbon-centered radicals located randomly at the polymer chains are the basis of the described method of nanogel synthesis.

## 5. Synthesis Overview

The reaction starts when aqueous solutions of a hydrophilic polymer are exposed to ionizing radiation of a sufficiently high dose rate. This creates many radicals along the polymer chain. Pulse irradiation by high-energy electrons from an electron accelerator is often the source of radiation used. 

The polymer radicals are relatively short-lived and can undergo various reactions. Figure 2 shows a general scheme of the most common reactions. Radical recombination reactions result in the formation of new C–C bonds in aliphatic polymers. The two recombination reactions of most interest are intermolecular crosslinking (shown in Figure 2c) and intramolecular crosslinking (Figure 2d). Intermolecular recombination reaction takes place between two radicals on separate macromolecules while intramolecular recombination reactions take place between two radicals on the same chain. Intramolecular crosslinking is sometimes called loop formation, and it is this reaction that is the basis for the synthesis of NGs. 

Intermolecular crosslinking creates high molar mass species by linking many chains together, and further, beyond the gelation dose, leads to the onset of a 3D macroscopic gel. This is a standard way to synthesize hydrogels using ionizing radiation, and also a base of the industrially employed technology of production of hydrogel wound dressings [55,56]. Intramolecular crosslinking, on the other hand, by inducing loop formation on single coiled polymer chains, creates collapsed three-dimensional internally crosslinked chains (referred to as nanogel particles) without an increase in molar mass [57]. In drug delivery, small particle size such as with these nanoparticles, is essential in transporting bioactive molecules to target sites.

Other types of reactions may also occur in a polymer irradiated in solutions. For example, two macroradicals may react and form C–C double bonds. This process is called a disproportionation reaction (Figure 2e,f). Here, no crosslink is formed except for a double bond formation at one of the parent radical sites. Disproportionation reactions can be inter- or intramolecular. Neither leads to nanogel synthesis because there is no polymer crosslinking involved. Macroradicals can also experience degradation in which bonds are broken at various locations of the polymer chain. Degradation (or chain scission) causes the polymer chain to break two chain fragments, with one containing a terminal radical (Figure 2a). H-atom shift or a radical transfer can also occur (Figure 2b) between short or distant chain segments. H-atom shifts have also been reported to occur between separate chains.

Radiation synthesis provides a means for controlling the competition between intermolecular and intramolecular crosslinking, thus allowing the production of macroscopic (“wall-to-wall”) gels or nanogels. The key parameter is the average momentary number of radicals per chain, *Z_R_*_0_. When this value is low (typically < 1), crosslinking takes place predominantly between single radicals located on separate macromolecules. On the other hand, simultaneous presence of multiple radicals on each chain (*Z_R_*_0_ >> 1) promotes intramolecular recombination. Low *Z_R_*_0_ values are typical for moderate dose rate continuous irradiation of polymer solutions at medium to high concentrations (Figure 3a). In contrast, when polymer concentration is low (much below the critical hydrodynamic concentration, so that the coils are separate and do not interpenetrate), it is easy to generate many radicals per chain using high-dose pulse irradiation such as a pulsed electron beam; here, predominantly intramolecular recombination leads to the formation of nanogels (Figure 3b). In practice, it is difficult to achieve pure inter- or intramolecular crosslinking, but the above-described approach allows the choice of which of these processes dominates. For a procedure that combines these approaches to produce nanogels or microgels of desired size and density, see ref. [58].

Temperature is another parameter that can influence the formation of crosslinks in NGs [24]. At elevated temperatures, the polymer chain is said to “collapse” as hydrogen bonding between the polymer and water molecules is destroyed. The collapsed chains are of a stiffer conformation and produce smaller NGs upon irradiation (Figure 4). This is due to a decrease in distance between the C-centered radicals on the same chain. This in turn favors intra-crosslinking. However, inter-crosslinking is still observed. This is because despite the closer proximity of radicals on the same chain, the higher temperature also increases the diffusive motion of chains, thus allowing more contacts and chances for inter-crosslinks to form. In these conditions, intra-crosslinking has shown to be dominant by the smaller dimensions of the final NGs. However, the increased molecular weights of these gels also suggest the presence of some inter-molecular crosslinking. The decrease in pervaded volume at higher temperature also implies less ‘empty’ space (empty here is referring to spaces where only solvent molecules are present). Large pervaded volumes increase the chance of the hydroxyl radicals decaying before interacting with the polymer. When this happens, hydroxyl radicals will often react with themselves to produce hydrogen peroxide, decreasing the radicals available to abstract hydrogens from the polymeric backbones [60,61].

Disproportionation reactions often compete with recombination reactions because both occur in similar yields, while for some polymers, like poly(vinyl alcohol), it has been shown that disproportionation dominates over crosslinking [62]. This has to be taken into consideration when planning the syntheses of gels, both macroscopic and nano-sized. Chain scission or H-atom shift reactions can influence the kinetics of radical decay, but do not actually change the number of radicals in the system. Chain scission can, for example, lead to shorter chains, which may cause faster intermolecular reactions which would not be ideal for nanogel synthesis.

While in this review will focus on nanogel synthesis by radiation-induced intramolecular crosslinking of polymer chains, it should be noted that there are other approaches where nanogels or microgels can be obtained using ionizing radiation. The simplest of these is based on crosslinking polymerization in bulk, resulting in solid, macroscopic crosslinked polymer material. This product is then ground in order to obtain microgels; nanogels are hard to make in this way [63]. Micelle-forming polymers such as graft copolymers of polyNIPAAm and 2-alkyl-oxazoline can be crosslinked in the micellar state, yielding core-shell nanogels [64,65,66]. A range of gel particles of various dimensions and some reactivity to pH and temperature have been obtained by gamma irradiation of gelatin solutions [67]. Irradiation of monomers and/or crosslinking agents in inverse micelles formed in *n*-heptane/water/surfactant system has been reported to yield nanogels both when electron beam or gamma rays were applied [68,69]. An interesting idea for synthesizing PAA/PVP nanogels is to use radiation-induced template polymerization of acrylic acid on PVP chains in aqueous solutions [69,70,71,72]. Such products have been tested for controlled release of some model drugs.

## 6. Kinetics

### 6.1. Decay Kinetics and Mechanisms of Polymer-Derived Radicals

Intramolecular recombination of radicals within a flexible polymer chain has interesting kinetic features. While it has some general characteristics of a second-order reaction, as is typical for intermolecular radical–radical recombination in low molecular weight compounds (namely, if all other conditions are constant, the half-life becomes shorter when the initial radical concentration is increased), detailed analysis indicates that a homogeneous second-order kinetics model is not capable of describing these reactions. The 1/c vs. time plot, where c is the momentary radical concentration, is no longer linear but bent down. The momentary apparent rate constant decreases with reaction time (see examples in Figure 5).

In an early study on radiolysis of poly(ethylene oxide) (PEO) solutions [73] it was found that, under conditions where intramolecular recombination prevails, increasing polymer concentration while keeping the initial radical concentration constant causes the reaction to slow down (Figure 6a). It has been observed that the reaction rate was actually governed neither by polymer concentration nor by radical concentration alone, but by the initial average number of radicals per chain, *Z_R_*_0_ (Figure 6b).

Moreover, it has been found that the kinetic traces couldn’t be well described using the classical second order kinetics (Figure 7a), while a much better fit has been observed when a dispersive kinetics model was used, such as the one proposed by Plonka [74,75] where the momentary value of the rate constant *k*(t) changes in time:*k*(t) = *B t*^α−1^(5)
where *t* is time, *B* is a constant, and α is a parameter attaining values between 0 and 1 which can be used to indicate how far the observed process deviates from the classical kinetics; for α = 1 the rate constant does not depend on time, so the kinetics is classical, while lower values of α indicate non-classical kinetic behavior. The above-mentioned study on PEO indicates that α is significantly lower than one when the average number of radicals per chain, *Z_R_*_0_, is much higher than one. On the contrary, for *Z_R_*_0_ << 1, classical kinetics with α ≈ 1 was observed (Figure 7b). It should be noted that in these experiments *Z_R0_* was adjusted not only by varying the dose per pulse, but also the chain length, thus the data for *Z_R_*_0_ << 1 originate from PEO oligomers. Whether or not at *Z_R0_* << 1 radicals located on long polymer chains would yield α ≈ 1 remains open.

The dispersive kinetics model stipulates that the reactivity distribution changes over time. The reason why this should happen in the process of intramolecular crosslinking became clear when Monte Carlo simulations (Cooperative Motion Algorithm, CMA) were employed [76]. It turns out that if only two radicals are generated on a polymer chain and the distance between these two radicals, *N_R_*, is constant, their recombination follows classical kinetics and the rate depends almost solely on *N_R_*, with only slight effect of the polymer chain length *N* (Figure 8a). For random initial radical distribution along the chain, the average initial *N_R_* value is the main factor controlling the kinetics. Figure 8b illustrates the fact that kinetics of radical recombination on chains of different length *N* is similar if the average inter-radical distance along the chain, *N_R_*, approximated as *N*/*Z_R_*_0_, is constant.

In general, the momentary rate constant depends on the average inter-radical distance along the chain at a given time (*t*). Since on a flexible chain the closely located radical pairs react first, the distribution of inter-radical distances changes over time, thus affecting the temporary rate constant. This is a characteristic feature of dispersive kinetics. Should an efficient mechanism exist providing the possibility of efficiently restoring the distance distribution during the decay process, the departure from classical kinetics would be less pronounced. In fact, H-shift reactions, effectively allowing the radical site to change its location along the polymer chain, have been observed in a number of water-soluble polymers [62,77]; however, these first-order reactions are typically too slow to significantly influence the intramolecular recombination process. In some cases, when the structure of the polymer coil is prone to temperature-induced changes, such temperature transition is reflected in a change of the activation energy of radical recombination [22]. The dispersive kinetics of intramolecular recombination have been observed in many cases by experiments and simulations, and it seems to be a typical feature of these reactions [58,62,73,76,78,79,80,81,82].

### 6.2. Pulse Radiolysis

Pulse radiolysis is one of the classical methods for studying the kinetics and mechanisms of fast chemical reactions [83,84]. It is based on subjecting the studied system, usually a solution of substrates, to a short pulse of ionizing radiation in the form of high-energy electrons from an accelerator, and subsequently following the radiation-induced processes by a suitable time-resolved detection technique. While current advanced systems allow for pulses and detection resolution in the sub-picosecond time range [85], nanosecond time scales are usually sufficient for following the reactions of polymer radicals. Various detection techniques are used in pulse radiolysis to detect the short-lived transient species (or the decay of a substrate, formation of a final product, or change in some physicochemical property of the studied system); however, spectrophotometry is by far the most typical one. A simple setup for nanosecond pulse radiolysis is shown in Figure 9a. The sample is located in a quartz cell. A beam of analyzing light (white light from a lamp or monochromatic light from a laser) passes through the cell and the intensity of transmitted light vs. time is detected. A short electron pulse applied to the sample induces chemical reactions, which manifest themselves as changes in absorbance at the analyzed wavelength or, more generally, as changes in the spectral properties of the system. Analyzing the spectra of transient species leads to their identification, and this information coupled with kinetic data on the formation, transformations and decay of transient species allow for elucidation of the reaction mechanism. Thanks to this technique, thousands of rate constants of fast chemical reactions have been determined, and the radiation chemistry of many important compounds, including biomolecules such as DNA and proteins, has been described. Classical pulse radiolysis with spectrophotometric detection is also very useful in studying the free radical chemistry of polymers; however, for investigating some specific aspects of polymer systems, e.g., reactions leading to changes in molecular weight and molecular dimensions, pulse radiolysis with other detection techniques such as conductivity [86,87,88] and light scattering [89,90,91,92], has proven to be valuable.

While radiation synthesis of nanogels can be performed on various experimental setups, including an industrial electron beam processing line, for small to medium scale synthesis a system similar to those used for pulse radiolysis has proven to be very useful. This is since it allows for precise control of parameters and uses much lower doses than industrial systems (thus avoiding unwanted secondary side reactions), continuous deoxygenation of the solutions, and collection of samples during the process (Figure 9b) [59]. Polymer solutions are saturated with argon or N_2_O in the solution reservoir, circulating in a closed loop passing through a cell that is pulse-irradiated at a given frequency, where the dose per pulse can be as high as ca. 1 kGy. Knowing the volumes of the cell and the reservoir, one can calculate the average dose absorbed by every volume element of the solutions. Typically, synthesis requires running the solutions through a few irradiation cycles when the accumulated dose corresponds to the absorption of several electron pulses by each volume element of the liquid. An obvious advantage of this system over, e.g., gamma irradiation is that in the irradiated volume the momentary radical concentration is high, and thus the average number of radicals per polymer chain can be much higher than one, which is a prerequisite for efficient intramolecular crosslinking.

### 6.3. The Importance of the Stockes–Einstein Equation to Prove Intra-Crosslinking Reactions with the Single Chain

A typical example of this case is the pulse radiolysis experiment of N_2_O-saturated aqueous solutions of PVP. Figure 10 [22] shows the second-order decay kinetics of the C-centered radicals.

To validate the assumption that the PVP^•^ decay represents an intracrosslinking reaction, it is important to compare the estimated collision time with the observed decay time of the radicals. This can be done by considering the diffusivity data and the Stokes–Einstein equation in order to determine the collision time of two separate polymer chains. Under these experimental conditions shown in Figure 10, based on the calculations below the estimated collision time of two PVP^•^ radicals is at least five times longer than the observed decay time in Figure 10. The calculation of the collision time can be determined as follows (1) based on Figure 10; the initial transient absorbance of the PVP^•^ is *A_0_* = 0.045. Therefore, the concentration of the [PVP^•^]_t=0_ at the start of the decay is *A_0_**/**ε l* = 0.045/(1082 L mol^−1^ 2 cm) = 2.08 × 10^−5^ M, *(l =* pathlength). Therefore, the average number of radicals per polymer is 8.2. In order to determine whether the second order decay of the C-centered radicals relates to inter-crosslinking or intra-crosslinking, the measured decay time needs to be compared with the determined average time between collision of two polymer radicals. By applying the dynamic light scattering method, the diffusivity, *D_t_*, can be measured and found to be 1.0 × 10^−11^ m^2^ s^−1^ at room temperature. Using equation *t* = *x*^2^/6 *D_t_* [22], where *x* is the mean free path, the mean collision time was found to be 120 µs at 20 °C and 0.2 ms at 77 °C, assuming that the C-centered radicals are uniformly distributed throughout the solution. The typical half-time observed in our pulse radiolysis studies is perhaps a tenth of this value. These results indicate that intramolecular second order kinetics is appropriate for the analysis of the pulse radiolysis measurements.

### 6.4. Considerations for Dilute PVP Aqueous Solutions

In this case, the appropriate C to use in the polymeric system is the effective molar concentration, C_r_, of the monomer units in a polymer molecule occupying a pervaded volume (sometimes called hydrodynamic volume) $V_{h}$ = ($\frac{4}{3}$)Π$R_{h}^{3}$, where R_h_ is the hydrodynamic radius. To validate this assumption, it is necessary to assume that the polymer chains are not aggregated (or, almost not aggregated), and that they are separated during the pulse.

The spatial distribution of polymer molecules in the dilute aqueous medium can be described in terms of a random distribution model [93], which is based on the Poisson distribution of *Z* molecules in *N* cells:
(6)Pm = ZNme−ZNm!
where *P*(*m*) is the probability that a cell contains exactly m polymer molecules.

The number of PVP molecules in a liter of solution *Z* was calculated as 1.53 × 10^18^ and the imaginary number of cells *N* which can accommodate isolated polymer molecules was estimated as 1.96 × 10^19^, as shown in the previous work [22]. According to this equation, *P*(0), the probability that a cell remains unoccupied is:*P*(0) = *e*^−Z/N^ = *e*^−0.0779^ = 0.9251(7)

It also follows from Equation (6) that *P*(1), the probability that a cell is occupied by one polymer molecule, is (*Z/N*)⋅*e^−Z/N^* = 0.0721, while *P*(2) = (*Z/N*)⋅(*e^−Z/N^/2*) = 0.0028, *P*(3) = 0.00007 and the probability that more than one cell is occupied by three or more polymer molecules is negligible. This calculation shows that the number of cells occupied by one polymer molecule, *P*(1), is 0.0721*N*, while the total number of occupied cells, [*P*(1) + *P*(2) + *P*(3) + *P*(*m* > 3)]*N*, is [1-*P*(0)]*N* = (1-0.9251)*N* = 0.0749 *N*. Thus, no more than 1-0.0721/0.0749 = 0.038 or 3.8% of the occupied cells are occupied by more than one molecule. Considerations of excluded volume and of chain geometry reduce the number of cells occupied by more than a single polymer molecule even further.

### 6.5. The Role of Dispersive Kinetics (Plonka’s Model)

Experimental data in some studies of the decay of radicals produced by irradiation of polymeric systems have shown that there are changes in the rate constant of bimolecular recombination reactions with time. This finding has been interpreted using the dispersive kinetics model [74],which assumes that due to the distribution of distances among the radicals those that are formed closer to each other will recombine more rapidly than those that are separated by a larger distance along the backbone of the chain. Plonka’s equation describes the time-dependent nature of dispersive kinetics (Equation (5), see above) [75].

In Equation (5), *B* and *α* are both constant at a given temperature. The constant *B* is temperature-dependent. The power *α*-1 represents the impact of the dispersive effect, i.e., in cases where *α << 1*, the dispersive effect, i.e., the dependence of the rate constant on time, is very strong, while in cases where *α*
*≈ 1* this effect is not significant. In our work [22], fitting the experimental data to Equation (5) yields values of *B* = 2860 and *α* = 1.135. This value of *α* is close enough to 1 to indicate that the dispersive effect is not highly significant, and it is consistent with the observation that the data are consistent with a homogeneous second-order behavior. This can be attributed to the fact in the present experiments, which were performed using a moderate dose per pulse, only about 8.2 radicals are produced per polymeric chain containing 3540 ± 100 monomeric VP units in a single pulse (see above). This might not be the case in studies employing much higher doses per pulse. 

### 6.6. The Effects of Chain Conformation on the Decay Kinetics of PVP^•^ Radicals

Figure 11a shows selective data presenting the effects of the temperature on the second order kinetics decay of PVP^•^, and Figure 11b shows the Arrhenius plots of these data.

As shown from Figure 11a, the bimolecular decay of the (2*k*_2_) values rise much more steeply with increasing temperature in the high temperature range (55 °C to 77 °C) than in the low temperature range (28 °C to 50 °C). As a result, the Arrhenius plot reveals two activation energies, as depicted in Figure 11b. The data for the high temperature region (above 55 °C) yields a higher *E_a_* of (28.5 ± 7.9) kJ mol^−1^ than the data for temperatures below 55 °C (4.2 ± 1.7) kJ mol^−1^.

In the higher range of temperatures (55 °C to 77 °C), the mean distances between radiolytically generated carbon-centered radicals on the backbone of a single chain are expected to be shorter than those at lower temperatures (28 °C to 50 °C). This is due to the collapsed chain conformation at higher temperatures due to the destruction of the hydrogen bond between PVP and H_2_O. The destruction of the hydrogen bonds between PVP and H_2_O renders the chains to be stiffer.

Therefore, higher 2*k*_2_ values as observed in the high temperature region are not unexpected. However, at low temperatures the polymer chain forms a non-collapsed random coil, with dimensions which are closer to those predicted by random walk statistics, since the intra-chain interactions are weak. Therefore, at the lower temperatures the mean radical–radical distance on a single chain is longer, and in this case lower 2*k*_2_ values are observed.

This is explained by the change of polymer chain conformation and the different rate-determining mechanisms of the PVP^•^ radical recombination reactions in each temperature range. The change in activation energy around 55 °C matches the decrease in *R_h_* and the corresponding increase in *D_t_* that take place at this temperature range according to the Asymmetric Flow Field Flow Fractionation, AFFFF or AF4 measurements, as shown in Figure 12.

In the higher temperature range, the reduced hydrodynamic interactions give way to hindered movement of the radical-bearing chain segments within the collapsed coil due to the increased friction within the relative solid-concentrated (i.e., solvent-depleted) region, leading to a higher *E_a_*. Of course, the rate of inter-molecular recombination increases with temperature in the higher temperature region (above 50 °C–55 °C) as well as in the lower temperature range. However, in the higher temperature range this increase is overshadowed by the fact that intra-molecular recombination is the dominant mechanism in the collapsed coils. This change in mechanism is correlated with the much higher activation energy observed in the higher temperature range.

The published data from various labs show how the radiolytically produced ^•^OH radicals from the radiolysis of H_2_O can abstract H atoms from the backbone of the chains, and also to lesser extent from the functional groups along the chains. The locations of the C-centered free radicals produced from the abstraction reactions depend on the neighboring atoms, such as oxygen and nitrogen. For example, ^•^OH radicals abstract H-atoms from the ethyl group and the pyrrolidine group along the backbone of the PVP. Due to the weak C-H bond of the tertiary carbon (-CH-) in the ethyl group, ^•^OH radicals can easily abstract H-atoms (-C-H-) with a reaction rate constant of ~2 × 10^9^ L mol^−1^ s^−1^. On the other hand, owing to the stronger C-H bond of the secondary carbon (-CH_2_-) atom of the ethyl group, ^•^OH radicals abstract the H-atom on the ethyl group with much less reactivity at reaction rate constant of ~10^8^ L mol^−1^ s^−1^ [94,95]. In addition, due to N-atom-induced activation ^•^OH radicals concurrently abstract H-atoms from C4 on the pyrrolidine group. It has been reported that while the abstraction of H-atoms from C4 of the 2-pyrrolidone-2-one is 2.2 × 10^9^ L mol^−1^ s^−1^, the overall abstraction of H from *N*-ethylpyrrolidone (from the ethyl and the pyrrolidone groups) is 2.9 × 10^9^ L mol^−1^ s^−1^ [96]. Based on these observations, one could estimate that in the case of the PVP, the probabilities of abstracting H atoms by ^•^OH from the ethyl and the C4 of the pyrrolidone groups are almost equal.

## 7. Product Analysis

### 7.1. Classical Methods

In experiments on the synthesis of nanogels by radiation-induced intramolecular crosslinking of polymers, reaction progress and formation of nanogels can be followed by a number of techniques. The most characteristic feature of this process is a decrease in the macromolecule’s radius of gyration (*R_g_*) even when the average molecular weight may remain consistent or increase (the increase being due to the fact that usually there is some contribution of intermolecular recombination). Since both *R_g_* and weight-average molecular weight can be determined simultaneously by the multi-angle static laser light scattering technique, MALLS, and *M_w_* is determined by this technique in an absolute way (with no calibration needed), MALLS can be considered the method of choice for tracking nanogel formation. Actually, one may use these two parameters to estimate a coil density which is understood as polymer mass divided by the volume it occupies. MALLS is also valuable in studying more complex systems, where crosslinking is accompanied by side reactions such as chain breakage.

Viscometry can be used as an auxiliary technique; the increasing compactness of polymer coils due to intramolecular crosslinking leads to a drop in viscosity. One should bear in mind, however, that while viscometry can be applied for linear chains to determine the (viscosity-average) molecular weight of a polymer using the Mark–Houwink equation, the parameters of this equation do not hold for internally crosslinked samples; therefore, in most cases the viscometric approach cannot be applied to follow molecular weight changes in the course of nanogel formation. Determination of hydrodynamic radius (*R_h_*) can also be helpful; however, due to the transformation of topology of macromolecules in the course of the synthesis, *R_h_* changes are usually less pronounced that those of *R_g_*. Nevertheless, the *R_g_*/*R_h_* ratio is a useful tool for following the changes in macromolecular architecture from loose coil towards a less penetrable and more compact sphere.

Modern gel permeation chromatography (HPLC-GPC) equipped with refractometric, viscometric and MALLS detectors may be a useful tool in further studies on nanogel syntheses [60,97,98], while its column-free version, AF4 (see discussion below), can also be of interest when the intended products are of a somewhat larger size (microgels) [22]. In addition, the application of nanoparticle tracking analysis [99,100,101] in these studies is of potential interest.

When working with polyelectrolytes, it may be of interest to know the zeta potential of the resulting nanogels, a parameter which may help in assessing the electrostatic stability of the products and its interactions with other ions, molecules and surfaces.

Visualization of individual nanogels is a difficult task, mainly due to the soft and “empty” nature of these products. Usually, the first step in visualization is drying, which may cause the nanogels to flatten out. Even then, one can differentiate between a relatively flat surface obtained by drying the solutions of the linear parent polymer and the “hilly” surface resulting from drying a nanogel solution [22,59]. Optimization of sample preparations and visualization conditions leads to good quality AFM and SEM pictures of nanogels [97,102].

Monte Carlo simulations, as the above-mentioned CMA, can also be useful to visualize and analyze the structure of nanogels as well as the influence of various factors (substrate properties, irradiation conditions) on the architecture of the final products [76].

While intramolecular crosslinking alone is not expected to bring about significant changes in the chemical structure of macromolecules other than the formation of crosslinking bonds, one should remember that, in practice, side reactions may occur. One of them is disproportionation of radicals, which competes with crosslinking (in some cases, as in poly(vinyl alcohol), very efficiently [62]) and leads to the formation of double C=C bonds. Of course, following the chemical structure of the product is even more important if additional substrates (for instance, monomers) have been present in the system during the synthesis. Such an approach has been used by Dispenza et al. to obtain PVP-based nanogels containing some carboxylic groups; for this purpose, moderate amounts of acrylic acid were added to solutions of linear PVP substrate [25,103,104]. Another aspect worth mentioning is that while typical nanogel synthesis by radiation-induced intramolecular crosslinking is performed in oxygen-free conditions, incomplete scavenging of ^•^OH radicals by macromolecules (typically present at low concentrations) leads to the formation of hydrogen peroxide and, in consequence, at high doses may result in formation of oxygen and related side products [61].

### 7.2. Asymmetric Flow Field Flow Fractionation (AF4 or AFFFF)

Aside from the above-mentioned traditional characterization techniques, Asymmetric Flow Field Flow Fractionation (AF4 or AFFFF) is a new and interesting technique that can be used to separate polymers, nanoparticles and many biomolecules by size. This technique is unique from other chromatographic set-ups in that there is no need for a stationary phase. This eliminates concerns about how samples could potentially interact and change with a given stationary phase. Rather, AF4 is a single-phased liquid chromatography technique that relies on a field force applied laminar flow [105]. AF4 can be coupled with a UV-vis spectrophotometer and/or a Multi-angle Light Scattering detector to determine molecular weight information. Specifically, for NG characterization, the *R_h_* and molar mass of our material can be determined using AF4. Figure 13 illustrates the basic principle of field flow fractionation.

There are two phenomena or forces at play that make up the AF4 technique. The first consists of a laminar flow force along the channel. This parabolic flow pushes on the sample molecules and carries them forward at different velocities. The second force is an external separation field or cross flow that pushes the solute molecules to one wall in the channel. Diffusion through the Brownian motion of the particles creates a counter motion to the cross flow. Thus, heavier molecules stay near the bottom while the lighter molecules are able to diffuse up. Putting these two forces together, the smaller molecules end up moving faster down the channel and being positioned higher up. Larger molecules move slower and are positioned lower of the channel.

Note that since the height distribution within the channel depends on a particle’s ability to diffuse (i.e., its translational diffusion coefficient *D_t_*), it can be related to the hydrodynamic radius using the Stokes–Einstein equation.

More information about the underlying principles and equations related to Field-Flow Fractionation can be found in the literature [105,106].

## 8. Radiation-Induced Synthesis of Select Nanogels

### 8.1. Radiation-Induced Synthesis of Poly(acrylic Acid) Nanogels

The radiation chemistry of poly(acrylic acid) in aqueous solutions has been studied by pulse radiolysis and by analyzing final products, both for the polymer itself and using a model compound [77,108,109,110]. Since PAA is a weak polyelectrolyte (p*K_a_* for long chains is ca. 6.0 [111,112]), the fate of radiation-generated polymer radicals is highly dependent on pH. In alkaline and neutral solutions, strong coulombic repulsive forces acting between the chain segments cause the macromolecules to attain a rod-like conformation and impede recombination of radiation-generated radicals. In extreme cases, these radicals, otherwise not stabilized by any structural resonances, can last in the aqueous solutions at room temperature for hours [113]. As a consequence, other slow reactions can effectively compete with radical recombination, such as degradation (chain scission), leading to a pronounced decrease in average molecular weight. In order to promote radical recombination, irradiation should therefore be performed at low pH, keeping the balance between reducing the charge density to a minimum [111] and avoiding aggregation of PAA macromolecules by hydrogen bonding [114]. It has been demonstrated that pulse-irradiation of dilute, argon-saturated PAA solutions at pH 2 by electron beam is an effective way to synthesize PAA-based nanogels by the technique discussed in this work [23,59,80]. The choice of polymer concentration and total absorbed dose is not trivial, since at least three reactions of interest can be seen taking place side-by-side: intramolecular and intermolecular crosslinking as well as degradation. By studying dependence of weight-average molecular weight and radius of gyration on these parameters (Figure 14), one can conclude that at very low concentrations chain scission initially dominates. However, with increasing dose, after some internal crosslinks have been formed, the macromolecules become less prone to degradation, since every segment becomes linked to the other ones by more than one bond. Using too high concentrations leads to a pronounced increase in *M_w_*, which is typically an unwanted effect if the goal is to synthesize nanogels. It has been observed that at the highest accumulated doses the reduction in *R_g_* begins levelling off, indicating less effective intramolecular crosslinking, probably due to already high crosslink density and reduced mobility of chain segments

It has also been shown that replacing a sequence of high-dose electron pulses with continuous gamma irradiation at a dose rate of ca. 0.15 Gy s^−1^ leads to predominance of intermolecular crosslinking, which may be utilized for synthesizing PAA microgels, i.e., gels having *R_g_* and *R_h_* of several hundred nanometers [23].

Nanogels containing carboxylic groups can also be synthesized using the discussed method by irradiating other systems than aqueous solutions of poly(acrylic acid); see below.

### 8.2. Radiation-Induced Synthesis of Polyvinylpyrrolidone Nanogels

Polyvinylpyrrolidone, or PVP, is a water-soluble polymer of proven biocompatibility and long track record of various medical applications, for instance as a plasma volume expander, binder in drug formulations, component of disinfecting preparations, and a hydrogel used in contact lenses and burn wound dressings. The latter, in the form of 3-4 mm thick transparent hydrogel sheets of various dimensions, are synthesized on an industrial scale using radiation technology [56,115,116].

Due to this longstanding excellent experience with medical applications of PVP, this polymer is also a natural candidate for synthesizing nanogels, which may be considered as potential drug nanocarriers (see below).

Early studies on the radiation chemistry of PVP, in particular in the conditions of pulse irradiation, were performed by Davis et al. [96] and by Rosiak et al. [117], indicating the possible radical structures and confirming the presence of fast and efficient radical recombination processes upon irradiation of oxygen-free PVP solutions.

Ulański and Rosiak have demonstrated that pulse-irradiation of dilute, oxygen-free aqueous PVP solutions under carefully chosen conditions (concentration, dose per pulse) leads to considerable reduction in the macromolecular dimensions (*R_g_*), while at the same time weight-average molecular weight remains nearly constant, which is evidence of intramolecular crosslinking and nanogel formation [118].

Another study conducted by An and colleagues investigated the radiation-induced synthesis of polyvinylpyrrolidone (PVP) nanogels [22] (see also the section “The effects of the chain conformation on the decay kinetics of the PVP^•^ radicals” above). They irradiated dilute aqueous solution of PVP that was saturated with N_2_O with a pulsed electron beam. They aimed to maximize intramolecular crosslinking by increasing the number of carbon-centered free radicals on the polymer and preventing inter-molecular reactions. They were able to do this by using a high pulse repetition rate of electron beam irradiation. They also showed that inducing a collapsed PVP chain conformation at high temperature (77 °C) also favored intramolecular crosslinking reactions.

Dispenza et al. developed a method whereby PVP nanogels have been synthesized using an industrial electron accelerator working in a regular mode such as applied in large-scale radiation sterilization procedures [102,119]. This approach has been subsequently developed to include nanogels of PVP copolymers with aminopropyl methacrylamide [119,120] and with acrylic acid [25,103,104,121]. High total doses used in this technique have been shown to promote some degree of oxidation, which may be of advantage in further steps of modifying the product by functional molecules [60,61].

Kadłubowski et al. proposed and successfully tested on PVP an approach allowing independent control of the size and molecular weight of nanogels in radiation synthesis [58]. This procedure consisted of two steps. Initially, irradiation was performed in conditions promoting intermolecular crosslinking (high concentration and moderate dose rate). As a result, there was an increase in average molecular weight. When the desired molecular weight was reached, irradiation conditions were changed to those promoting intramolecular recombination (dilute solutions, high-dose pulse irradiation), where the molecular weight does not change significantly, while the dimensions were reduced until the designed *R_g_* or coil density was achieved.

Sütekin and Güven conducted a systematic study on the radiation synthesis of PVP nanogels and microgels using electron beam or gamma rays [97]. Products in a 30-250 nm size range were analyzed using a multitude of analytical techniques. The effects of polymer molecular weight, concentration, type of radiation source, dose rate and total absorbed dose on the nanogel size were described. This work also contained high quality SEM and AFM pictures of the obtained PVP nanogels.

Two recent reviews provide further details of nanogel synthesis, one of them being focused on products based on PVP [122,123].

In parallel to these developments focused on the synthesis of PVP nanogels, the kinetics and mechanism of PVP radiolysis in water has been the focus of many experimental and simulation studies. The kinetics of the ^•^OH reaction with PVP have been studied in some detail by pulse radiolysis employing the competition kinetics approach, indicating the effects of polymer molecular weight and concentration on the apparent values of the rate constants and pointing out that in dilute solutions the factor controlling the reaction rate is the size of polymer coil approximated as *R_g_* [124]. Jonsson et al. elaborated a numerical simulation model of the radiation chemistry of aqueous polymer solutions and analyzed over 50 different reaction conditions related to the formation and recombination of polymer radicals [61,82,125]. A recent careful pulse-radiolysis study by an international team allowed for pinpointing the structures of PVP radicals formed by ^•^OH attack and analysis of the competition between this reaction and self-combination of hydroxyl radicals, and made interesting observations regarding the recombination kinetics of PVP-derived radicals [126].

### 8.3. Radiation-Induced Synthesis of Binary Polymer Complexes; PAA and PVP

The functionality of nanogels is an important property for specific drug delivery purposes. The functionality of radiation-synthesized nanogels can be controlled in various ways. One can start the synthesis with a copolymer bearing appropriate chemical groups or perform the synthesis using polymer A and monomer B as substrates. Another method is to form interpolymer complexes and subsequently transform them into nanogels. Finally, functionalization can be achieved by chemically modifying nanogels after radiation synthesis. Some examples related to the synthesis of PVP-PAA nanogels are mentioned below. This combination is of particular interest due to the excellent biocompatibility and ease of radiation crosslinking of PVP as well as benefits of incorporating carboxylate groups of PAA (facile further coupling, e.g., to drugs or targeting moieties, electrostatic stabilization of nanogels).

Dispenza et al. have demonstrated that controlled amounts of carboxylic groups can be introduced during the synthesis to nanogels of polyvinylpyrrolidone by performing irradiation in the presence of certain amounts of acrylic acid [25,103,104]. It is worth mentioning that the same approach can be applied to introduce other functional groups to PVP nanogels [119,120].

It is also interesting that the same research group provided evidence that under specific synthetic conditions (high accumulated doses, 20–80 kGy) polyvinylpyrrolidone irradiated in water with no additives undergoes side reactions which give rise to minor amounts of free amino groups as well as carboxylic functions [60]. Radiation chemical yields of these groups are on the order of 1–3 nmol/J. As mentioned above, still another approach to synthesizing PAA/PVP nanogels by starting from PVP and acrylic acid is to use radiation-induced template polymerization of acrylic acid on PVP chains in aqueous solutions [69,70,71,72].

PAA-PVP nanogels can also be obtained by pulse-irradiating common solutions of these polymers after inducing formation of PAA-PVP hydrogen-bonded interpolymer complexes (IPC) by tuning the pH to the appropriate range [82,127,128]. Optimum acidity is necessary to provide uninterrupted sequences of many protonated carboxylic groups along the PAA chains (which requires pH to be much lower than the p*K_a_* of the polyacid, being ca. 6 [111,112]), and to prevent aggregation of the formed complexes, which may occur at pH < 2. Formation of nanogels consisting of covalently bound poly(acrylic acid) and polyvinylpyrrolidone chains is evidenced by the fact that by changing pH to neutral the substrates revert back to common solutions of independent chains, while in the irradiated systems the chains hold together despite destroying the hydrogen bonds. Ghaffarlou et al. published a comprehensive study on factors that affect the IPC formation between PVP and PAA and allow control of the properties of PVP-PAA based nanogels obtained by radiation crosslinking of these complexes [129]. The nanogels showed good colloidal stability and responded to changes in pH due to the presence of PAA in their structure. The size of the nanogels was smaller than the size of the IPC coil due to the formation of intra-chain crosslinks.

## 9. Possible Medical Applications in the Treatment of Alzheimer and Tumor Cells

Nanogels have many potential medical applications including the treatment of Alzheimer’s disease and for tumor targeting.

Recent studies have linked insulin resistance and insulin action to Alzheimer’s, as insulin and insulin receptors throughout the hippocampus and cerebral cortex play an important role in memory [26]. Insulin delivery to the brain presents many challenges. Nanogels, with their high biocompatibility and soft, rubbery consistency which mimics natural tissues, could potentially be the answer. Picone et al. studied a potential therapeutic application of an insulin nanogel-loaded system that was synthesized by high-energy irradiation of poly(vinylpyrrolidone-co-acrylic acid) [26,130]. The insulin was covalently attached to the nanogel (Figure 15). The biological characteristics of the nanocarrier include high biocompatibility and insulin protection against protease degradation. The insulin-conjugated nanogel was also able to bind to insulin receptors and trigger greater insulin activation signaling than free insulin. They also showed that the insulin-conjugated nanogel had great efficiency in transporting insulin across the Blood Brain Barrier (BBB). Firstly, this was demonstrated in vitro by creating a layer of mouse brain endothelial cells and proving that insulin transport across such a model BBB barrier is significantly facilitated by using nanogels as carriers [26]. Secondly, positive results were also obtained in vivo on mouse models, where binding insulin with nanogels led to enhanced brain delivery of insulin through the intranasal route [130].

Nanogels can also be functionalized with specific compounds that are able to link tumor cell surface molecules. Adamo et al. developed polyvinylpyrrolidone (PVP-co-acrylic acid) nanogels produced by pulsed electron beam irradiation which have been derivatized with monoclonal antibodies [131]. This functionalization resulted in faster cellular uptake and selective targeting. Another study concerned similar PVP-PAA nanogels derivatized with folate [25]. It has been demonstrated that these nanogels exhibited selectivity towards cancer cells. Additionally, a selected nanogel system was conjugated to Doxorubicin or the RNA specific for the Bcl-2 gene (Bcl-2 siRNA) through a redox-sensitive spacer in order to develop glutathione-responsive nanosystems, and also provided faster doxorubicin release at their destination, triggered by higher intracellular glutathione concentration. As evident from above, NGs have great potential in facilitating the target releasing of biological molecules in a controlled manner to target specific tumor sites.

Nanogels are also considered as carriers for nanoradiomedicine, where, after derivatization with particular peptides, they would be able to transport radionuclides to specific tumor tissues. Such systems may act as theranostic tools, both for cancer diagnostics and internal radiotherapy. In a recent study, Matusiak et al. successfully conjugated radiation-synthesized PAA nanogels with bombesin (aimed to provide selective binding to prostate tumor cells) and DOTA, a chelator for radionuclides [132]. It has been demonstrated that such constructs are capable of efficient and selective labeling, with ^177^Lu, providing a proof of concept for the radiation synthesis of nanogel-based radioisotope nanocarriers for theranostic applications.

## 10. Concluding Remarks

A tremendous amount of attention has been given to nanocarriers in drug delivery research. The high water absorption, good biocompatibility, and high stability of nanogels make them good agents for further research. Radiation-induced synthesis of nanogels promises efficient, non-toxic products that can be used in a variety of different medical applications. Specifically, they can be used to improve the efficiency of drug release at targeted sites of interest.

Low linear energy transfer (LET) ionizing radiation such as electron beams up to 12 MeV, as well as gamma rays, can be considered among the best methods for synthesizing hydrogels for drug delivery systems. The tremendous advantage of using ionizing radiation lies in the fact that no additives are added during the synthesis except the energy absorbed by the aqueous polymer solution. The radiation dose rate and temperature can affect the final average molar mass ($\bar{M}$_n_) of the polymers and the conformation of the irradiated polymer chains. These two factors, dose rate and irradiation temperature, have been used to achieve the desirable chemical structures and the physical properties of hydrogels for various drug delivery systems. Other factors such as polymer concentration, sometimes referred to the volume fraction, and the value of the initial $\bar{M}$_n_ can play a role in determining the chemical and physical properties of the radiolytically produced hydrogels.

The next step is the reactivities of those produced C-centered radicals, which mainly, in the absence of molecular oxygen, are the competing intra and inter crosslinking reactions. While the former produces many loops on the backbone of the chains, the latter leads to the formation of micro-hydrogels. The predominance of intra-crosslinking stems from the production of a greater number of C-centered radicals along the backbone of the chain per unit time, meaning high a dose rate and shorter distance between those free radicals along the backbone of the chain, indicating that dispersive kinetic decay has to take place. Furthermore, in order to enhance the probability of intra-crosslinking, higher temperatures can collapse the chains and render the distance between the C-centered radicals shorter. However, the diffusion coefficient increases proportionally with temperature, which enhances the inter-crosslinking reactions. Other important issues include the effects of temperature on the flexibility of the chains. By virtue of their chemical structures, all polymers used to synthesize hydrogels form hydrogen bonds with the solvent, namely H_2_O. The increase in temperature causes the destruction of these hydrogen bonds and ultimately leads to the collapse of the polymer chains. All published data shows clear decrease in hydrodynamic radius (*R_h_*) and the radius of gyration (*R_g_*) as the chain transformation occurs, from flexible with high end-to-end distance to stiff and collapsed chains with shorter end-to-end distance. Thus, it is feasible that at higher temperatures the collapsed chains will have three-dimensional globules. The transformation of the flexible chain to globules at higher temperatures due to the destruction of hydrogen bonds is a reversible process. Thus, ionizing radiation-induced intra-crosslinking reactions of the globules produces three-dimensional nanoparticles, which can be called nanogels. The same results can be achieved by delivering very high dose rates at room temperature.

In addition, since the chemical structures of some polymer chains used in synthesizing hydrogels consist of hydrophobic and hydrophilic components, the collapse of these chains at higher temperatures has a uniform pattern because the hydrophobic sections of the chain would be inside the globules while the hydrophilic parts would be on the outside facing the water. In the case of PVP, the ethylene groups of the chains would be inside the PVP globules, while the pyrrolidone groups are situated outside the globules, facing the water molecules. As a result, this increases the probability of the ^•^OH radicals abstracting hydrogen atoms from the pyrrolidone groups over the ethylene groups.

Another important factor is the initial $\bar{M}$_n_ of the polymers used to synthesize the desirable hydrogels. The diffusion of polymer radical chains is inversely proportional to their $\bar{\surd M}$_n_, which is well defined by Stokes–Einstein equation. Thus, to enhance intra-crosslinking, it is important to select higher $\bar{M}$_n_. On the other hand, free radical polymer chains with relatively lower $\bar{M}$_n_ diffuse much faster, and therefore the probability of their collision is higher, leading more inter-crosslinks to take place.

It is also crucial to take into account the effects of the initial concentration of the polymer aqueous solution on the kinetics. As shown in Figure 4, at low concentrations, every single chain has its own overlap volume fraction *φ** (here, *φ** is the volume fraction of a single molecule inside its pervaded volume), and the rest of the solutions are ‘empty’ (i.e., filled with solvent molecules such as water). At higher concentrations where *φ** = *φ* (volume fraction), the chains fill the empty space [133]. So, at low polymer concentrations, where *φ* < *φ**, intracrosslinking reactions are the dominant reactions. On the other hand, at higher concentrations where *φ* = *φ**, and *φ* > *φ**, the probability of intercrosslinking reactions increases.

In summary, in order to achieve the desirable radiation-induced formation of three dimensional nanosized granules, it is important to use a high dose rate, preferably high-dose pulse irradiation (enhancing Plonka’s conditions), low concentration, and a high molar mass of the polymers. For synthesizing nanogels from polymers which experience strong hydrogen bonding with water molecules, irradiation at elevated temperature is also recommended. This is due to the collapse of the polymer chains at high temperatures during the irradiation. As mentioned earlier, this collapse decreases the distance between the free radicals along the backbone of the chain. However, for polymers with lower critical solution temperatures (LCST) such as poly(vinyl methyl ether) and poly(N-isopropylacrylamide), both of which have LCST at ca. 34–37 °C, irradiation at high temperature increases the formation of the macrogel tremendously since the phase separation occurs at higher temperatures, which enhances intercrosslinking [134,135]. When the phase separation occurs at temperatures higher than LCST, it is expected that the ^•^OH radicals are not fully responsible for formation of the C-centered radicals along the backbone of the chains, since they are mainly produced and react in the aqueous phase. Accordingly, direct interaction of the polymer chains with ionizing radiation would be dominant, rather than indirect interaction through the radiolysis of H_2_O (production of ^•^OH). This would lead to different radiolytic intercrossing and other products. Therefore, for all practical purposes, in order to produce the three-dimensional nanosized globules (nanogels), the selected irradiation temperatures should be high enough to destroy the hydrogen bonds between the chains and H_2_O, but certainly should be lower than the LCST of the polymer used in order to avoid phase separation.

## Figures and Tables

**Figure 1 pharmaceutics-13-01765-f001:**
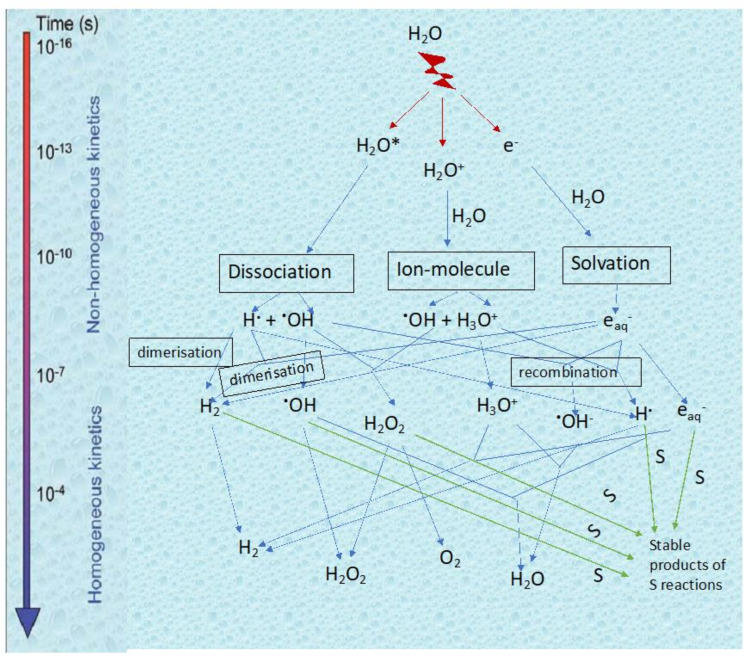
Scheme of reactions of transient species produced by irradiation in water without or with a diluted solute S acting as a radical scavenger. Adapted with permission from [50]; published by EDP Sciences, 2008.

**Figure 2 pharmaceutics-13-01765-f002:**
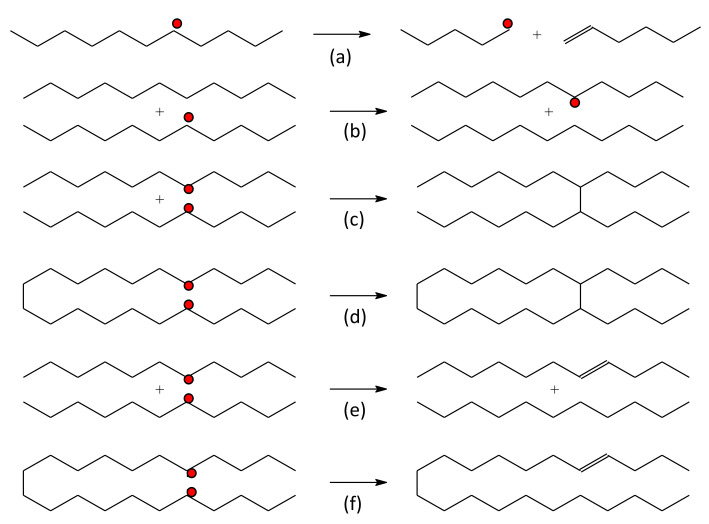
Schematic of general polymer radical reactions: (**a**) degradation reaction, (**b**) H-atom shift, (**c**) intermolecular crosslinking, (**d**) intramolecular crosslinking (basis of nanogel formation), (**e**) intermolecular disproportionation, (**f**) intramolecular disproportionation. Adapted with permission from [54]; published by International Atomic Energy Agency, 2016.

**Figure 3 pharmaceutics-13-01765-f003:**
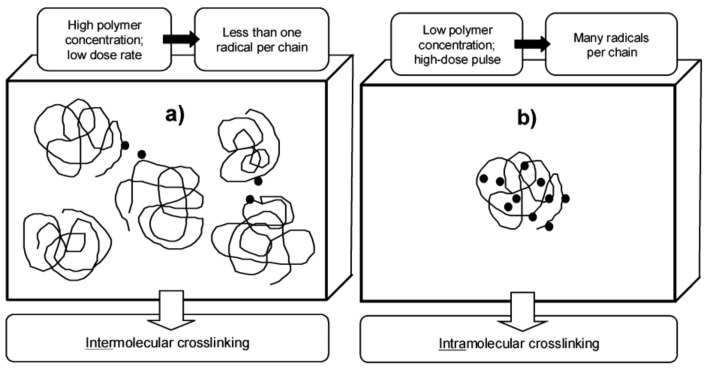
Irradiation conditions promoting (**a**) inter- and (**b**) intramolecular recombination of polymer-derived radicals in solutions. Adapted with permission from [59]; published by American Chemical Society, 2003.

**Figure 4 pharmaceutics-13-01765-f004:**
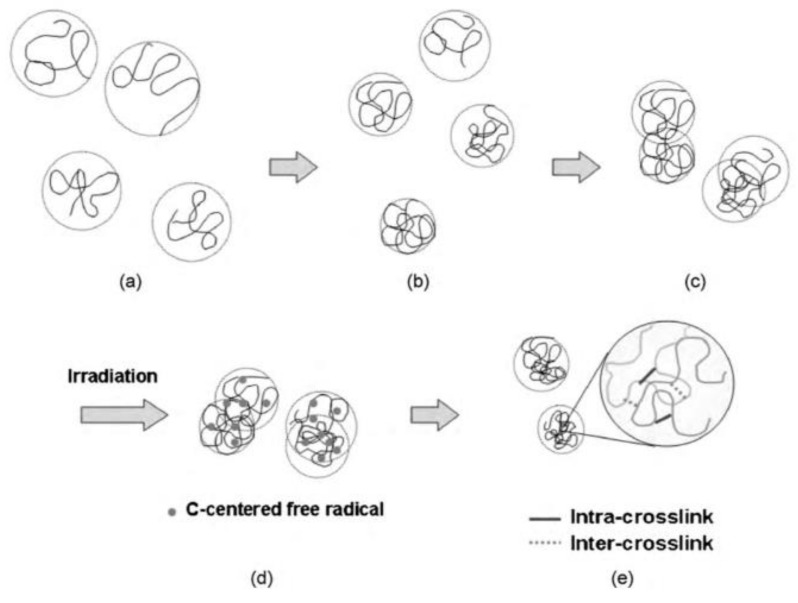
Synthesis of: (**a**) PVP NGs, dilute solution (c = 0.9 × 10^−3^ mol L^−1^); (**b**) High temperature, collapsed coils; (**c**) Diffusive interaction of coils; (**d**) C-centered radical formation; (**e**) Intra- and inter-crosslinks of PVP. Adapted with permission from [24]; published by Elsevier, 2010.

**Figure 5 pharmaceutics-13-01765-f005:**
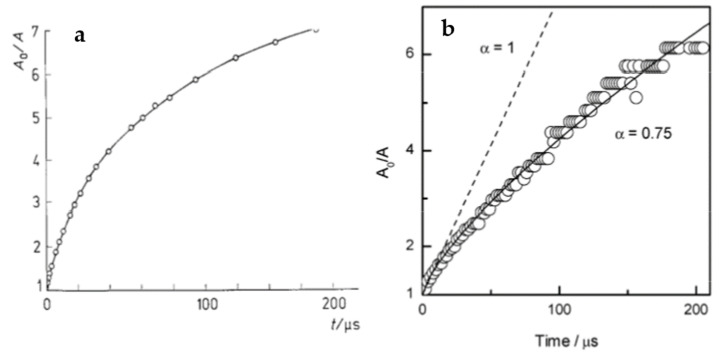
Exemplary second-order kinetic plots for intramolecular polymer radical recombination obtained by pulse radiolysis in deoxygenated, dilute aqueous solutions. (**a**) Poly(vinyl alcohol), *M_n_* ca. 55 kDa, polymer chain concentration 0.8 μM, initial radical concentration 22 μM, *Z_R_*_0_ = 28. Adapted with permission from [62]; published by John Wiley and Sons, 2003. (**b**) Poly(acrylic acid) at pH 2, *M_n_* ca. 260 kDa, polymer chain concentration 4.8 μM, initial radical concentration ca. 99 μM, *Z_R_*_0_ = 21. Adapted with permission from [59]; published by American Chemical Society, 2003. Dots denote experimental data; the broken line is the expected classical second-order behavior; the solid line is fitted using Equation (5).

**Figure 6 pharmaceutics-13-01765-f006:**
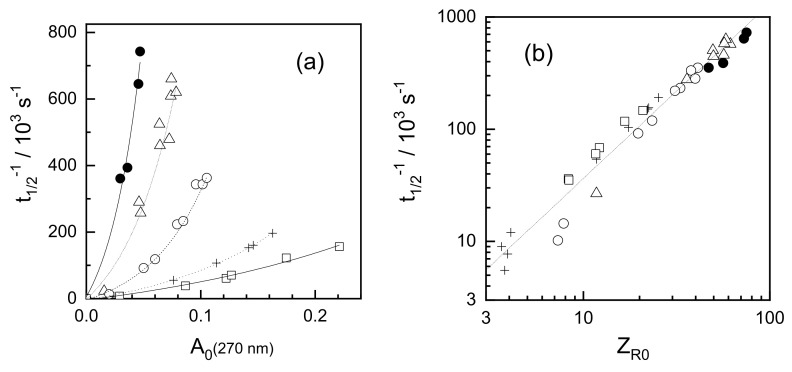
Pulse radiolysis of poly(ethylene oxide), *M_n_* = 218 kDa, in N_2_O-saturated aqueous solutions. PEO concentrations (in monomer units): 5.7 mM (●), 11 mM (△), 23 mM (◯), 57 mM (+) and 91 mM (☐). Rate of radical recombination (approximated as reverse half-life): (**a**) As a function of initial absorbance (proportional to the initial radical concentration); (**b**) As a function of the initial average number of radicals per chain (*Z_R_*_0_). Adapted with permission from [73]; published by Elsevier, 1995.

**Figure 7 pharmaceutics-13-01765-f007:**
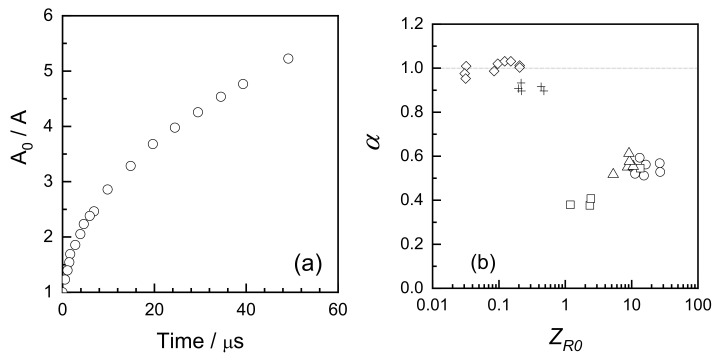
Pulse radiolysis of poly(ethylene oxide) in dilute N_2_O-saturated aqueous solutions: (**a**) Kinetics of radical recombination in the second-order kinetics coordinates (*M_n_* = 218 kDa, polymer chain concentration 2.2 μM, initial radical concentration ca. 120 μM, *Z_R0_* ca. 54); (**b**) Dependence of the α parameter (Equation (5)) on the average initial number of radicals per chain, *Z_R_*_0_. Data for PEO oligomer (*M_n_* = 400 Da) at oligomer chain concentrations of 110 μM (+) and 1100 μM (◇), as well as for PEO standards of *M_n_* = 23 kDa (polymer chain concentration 19 μM, ☐), *M_n_* = 94 kDa (polymer chain concentration 4.7 μM, △) and *M_n_* = 277 kDa (polymer chain concentration 1.6 μM, ◯). Adapted with permission from [73]; published by Elsevier, 1995.

**Figure 8 pharmaceutics-13-01765-f008:**
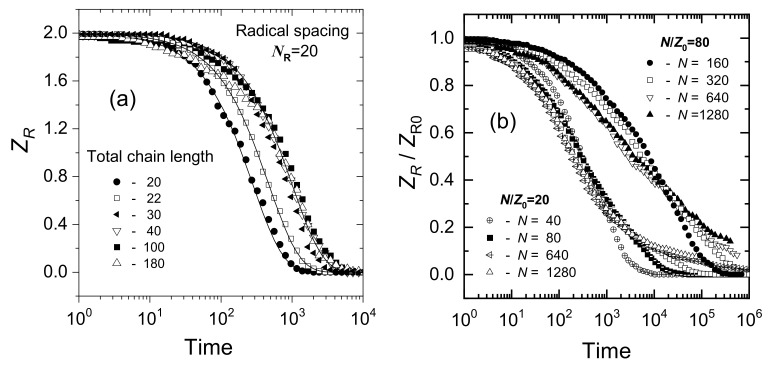
Monte Carlo simulation in intramolecular recombination of polymer radicals using the Cooperative Motion Algorithm. *Z_R_* denotes the momentary average number of radicals per chain; *Z_R_*_0_ is the initial average number of radicals per chain. Time is understood as a number of simulation steps: (**a**) Decay of single radical pairs (*Z_R_*_0_ = 2) generated at the fixed distance *N_R_* = 20 in the middle of chains of various length *N*; (**b**) Decay of radicals generated at random positions along the chain for two initial average interradical distances; *N_R_* ≈ *N*/*Z_R_*_0_ equal 20 and 80 for chains of different length *N*. Adapted with permission from [76]; published by American Chemical Society, 2006.

**Figure 9 pharmaceutics-13-01765-f009:**
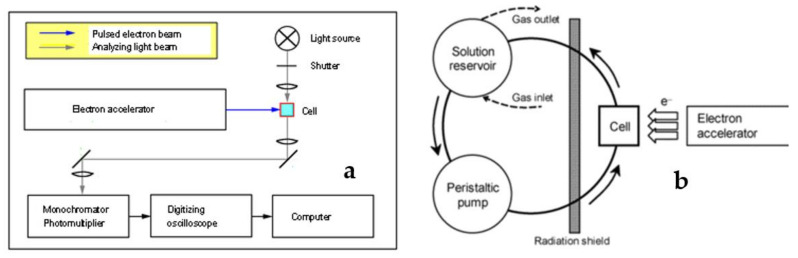
(**a**) Scheme of a simple pulse radiolysis setup with spectrophotometric detection, used in kinetic studies; (**b**) Scheme of a preparative pulse radiolysis system used for synthesizing nanogels. Adapted with permission from [59]; published by American Chemical Society, 2003.

**Figure 10 pharmaceutics-13-01765-f010:**
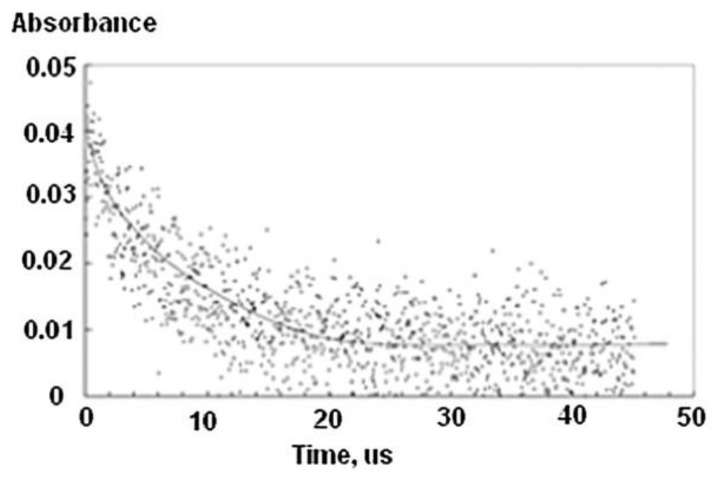
Decay profile kinetics at 390 nm of C-centered radicals in pulsed N_2_O-saturated aqueous solutions of 2.54 × 10^−6^ mol L^−1^ PVP (*M_w_* = (3.94 ± 0.12) × 10^5^ g mol^−1^) at 25 °C, pulse width 3 μs, dose per pulse 32 Gy. Adapted with permission from [22]; published by Elsevier, 2011.

**Figure 11 pharmaceutics-13-01765-f011:**
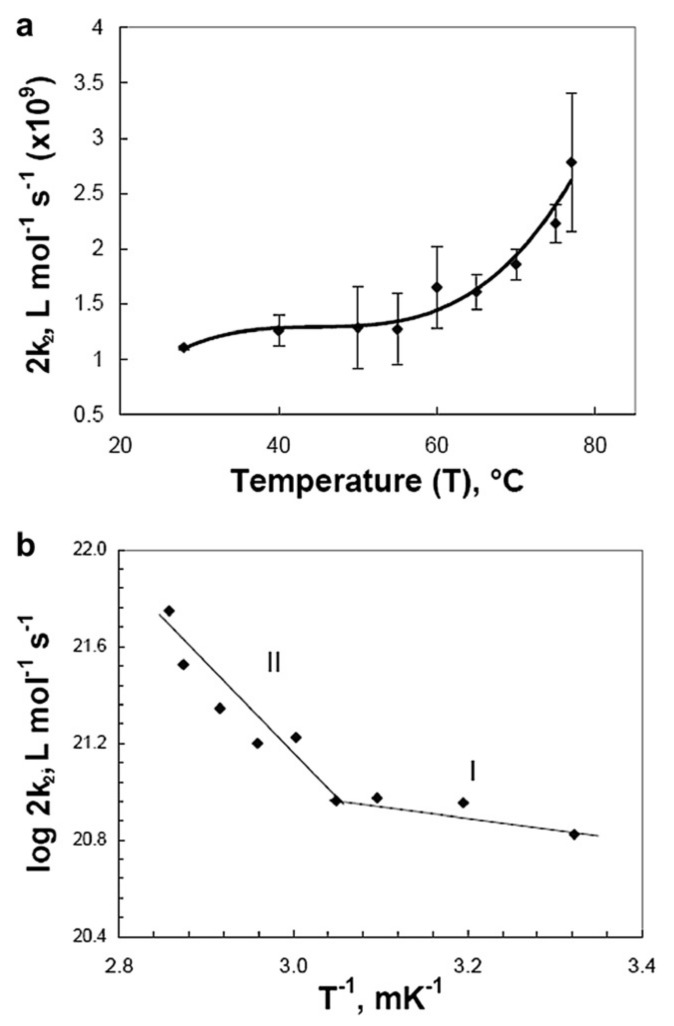
(**a**) Second-order reaction (decay) rate constant (2*k*_2_) as a function of temperature; (**b**) Arrhenius plot of *2k_2_* from which average activation energies *E_a_* = (4.2 ± 1.7) kJ mol^−1^ (I) and (28.5 ± 7.9) kJ mol^−1^ (II) were derived for two different temperature regions in N_2_O-saturated PVP aqueous solutions (2.54 × 10^−6^ mol L^−1^ PVP, *M_w_* = (3.94 ± 0.12) × 10^5^ g mol^−1^). Adapted with permission from [22]; published by Elsevier, 2011.

**Figure 12 pharmaceutics-13-01765-f012:**
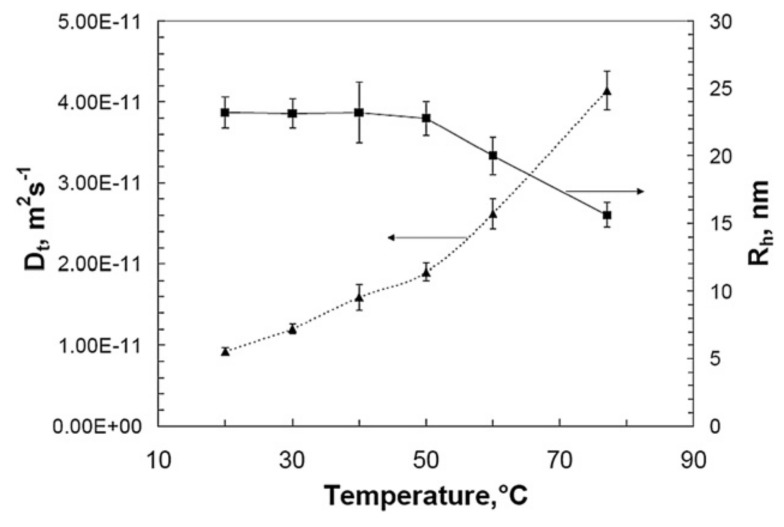
*D_t_* and *R_h_* of PVP polymer chains (*M_w_* = (3.94 ± 0.12) × 10^5^ g·mol^−1^) as a function of temperature. Adapted with permission from [22]; published by Elsevier, 2011.

**Figure 13 pharmaceutics-13-01765-f013:**
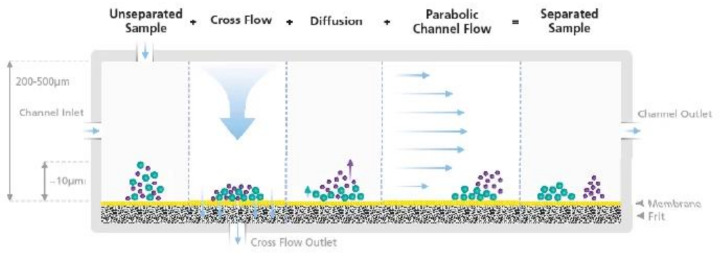
Operating principle of the AFFFF technique. Adapted with permission from [107]; published by LCGC, 2020.

**Figure 14 pharmaceutics-13-01765-f014:**
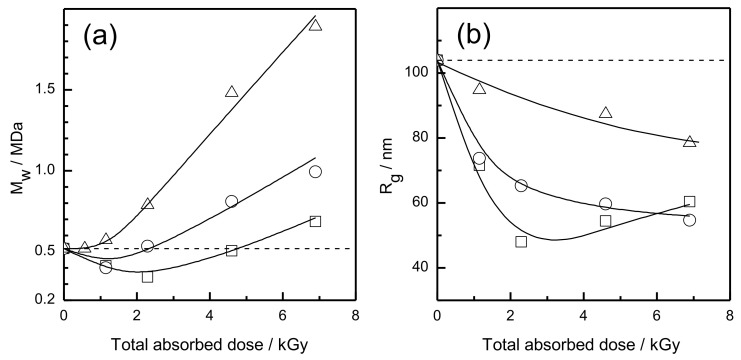
Changes in the weight-average molecular weight, *M_w_* (**a**), and in the radius of gyration, *R_g_* (**b**) of PAA molecules in the course of nanogel synthesis as a function of total absorbed dose (1.15 kGy corresponds to a single electron pulse) for samples of various PAA concentrations: ☐, 10 mM; ◯, 17.5 mM; △, 25 mM (in terms of the monomer units); irradiated in Ar-saturated aqueous solutions, pH 2. Radii of gyration measured at 25.0 °C in aqueous 0.5 M NaClO_4_, pH 10. Adapted with permission from [59]; published by American Chemical Society, 2003.

**Figure 15 pharmaceutics-13-01765-f015:**
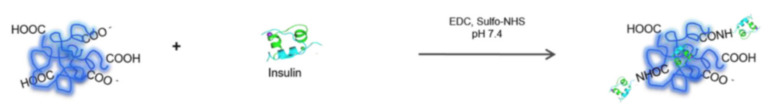
Conjunction of insulin with nanogels through amide bonds. Adapted with permission from [26]; published by Elsevier, 2016.

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
