# Peer review of "On the Mechanism and Kinetics of Synthesizing Polymer Nanogels by Ionizing Radiation-Induced Intramolecular Crosslinking of Macromolecules"

_pharmaceutics, 2021, doi:10.3390/pharmaceutics13111765_

Round 1

Reviewer 1 Report

The manuscript entitled «On the mechanism and kinetics of synthesizing polymer nanogels by ionizing-radiation-induced intramolecular crosslinking of macromolecules» submitted by Ashfaq et al. reviews in a methodical and very educational way the different synthesis steps and the characterization steps of nanogels as well as their potential in nanomedicine.

This review is well written and would be of great benefit to the scientific community.

A point remains however to be deepened on page 13, lines 861-863 "They also showed that the insulin conjugated nanogel had great efficiency in transporting insulin across an in-vitro and animal models of the Blood Brain Barrier (BBB)". Authors should develop this statement based on bibliographic references.

Author Response

To address the reviewers comment below:

A point remains however to be deepened on page 13, lines 861-863 "They also showed that the insulin conjugated nanogel had great efficiency in transporting insulin across an in-vitro and animal models of the Blood Brain Barrier (BBB)". Authors should develop this statement based on bibliographic references.

We have expanded on this section, please see below:

Nanogels with their high biocompatibility and soft, rubbery consistency which mimics natural tissues could potentially be the answer. Picone et al. studied a potential therapeutic application of insulin-nanogel-loaded system that was synthesized by high-energy irradiation of poly(vinylpyrrolidone-co-acrylic acid)[26], [130]. The insulin was covalently attached to the nanogel (Fig. 15). The biological characteristics of the nanocarrier include high biocompatibility and insulin protection against protease degradation. The insulin conjugated nanogel was also able to bind to insulin receptors and trigger a greater insulin activation signaling than free insulin. They also showed that the insulin conjugated nanogel had great efficiency in transporting insulin across the Blood Brain Barrier (BBB). Firstly, this has been demonstrated in vitro, by creating a layer of mouse brain endothelial cells and proving that insulin transport across such a model BBB barrier is significantly facilitated by using nanogels as carriers [26]. Secondly, positive results have been also obtained in vivo on mouse model, where binding insulin with nanogels led to enhanced brain delivery of insulin through the intranasal route [130].

We would also like to thank the reviewer for their time and efforts in reviewing our manuscript!

Reviewer 2 Report

This is a review about the synthesis of polymer nanogels by ionizing-radiation-induced crosslinking. The topicof the paper is of interest in nanomedicine. This review paper is well documented. The overall contents is well written and balanced in its parts.

Author Response

We would like to thank the reviewer for their time and effort in reviewing our manuscript!

Reviewer 3 Report

The manuscript involves a review of ionizing-radiation-induced intramolecular crosslinking of macromolecules for the formation of nanogels. The topic is timely and the manuscript is well organized and written. The scope o the review is quite broad involving a description of ionizing radiation and of water radiolysis, the possible  radical reactions for the formation of nanogels, kinetics, nanogel characterization methods, radiation induced synthesis of PAA and PVP nanogels, as well as possible medical applications thereof. Some subjects are discussed in great detail in the manuscript and I think this type of writing is more suitable for book writing. I suggest to revise carefully the manuscript in terms of coherency and specificity of the subjects in the field. Other review articles are also cited and the importance and difference of the present work in relation to the previous ones must be clearly stated.  

Author Response

To address this important point raised by the reviewer:

"I suggest to revise carefully the manuscript in terms of coherency and specificity of the subjects in the field. Other review articles are also cited and the importance and difference of the present work in relation to the previous ones must be clearly stated." 

We have expanded on our introductory section to better outline and introduce the contents to come with this paragraph:

This review aims to provide readers with the rudimentary information on the existing theories and practices of NG formation via. ionizing radiation. For this reason, the reviewer will cover a host of topics including a historical summary of water radiolysis, a synthetic overview of NG formation by radiation, the decay kinetics of the formed radicals and a range of product analysis techniques. Specifically the review will focus on the radiation-induced synthesis of polyvinylpyrrolidone [22], poly(acrylic acid)[23] and of interpolymer complexes of poly(acrylic acid) with polyvinylpyrrolidone [24] nanogels. Their applications for tumor targeting [25] will be discussed along with their use as insulin nanocarriers for a new type of treatment for Alzheimer’s disease[26]. Though a goal of this review is to be thorough, it is also the efforts of the writers to present the information in a concise and clear manner. This is so that experts and novices alike may benefit.  

We would like to thank the reviewer for their time and effort in reviewing our manuscript! 

Reviewer 4 Report

   The manuscript ID pharmaceutics-1389954, entitled "On the mechanism and kinetics of synthesizing polymer nanogels by ionizing-radiation-induced intramolecular crosslinking of macromolecules”, by authors: Aiysha Ashfaq, Jung-Chul An, Piotr UlaÅ„ski, Mohamad Al-Sheikhly, submitted to section: Nanomedicine and Nanotechnology, provide excellent, educative and informative review.

The abstract provides a satisfactory review of this manuscript.

In the Introduction part, authors have presented a short state of the art of nanogels. After that, eight sections follows: ionizing radiation; historical overview of water radiolysis; mechanisms of water radiolysis; synthesis overview; kinetics (decay kinetics and mechanisms of polymer-derived radicals, pulse radiolysis, the importance of Stockes-Einstein equation to prove intra-crosslinking reactions with the single chain, considerations for dilute PVP aqueous solutions, the role of dispersive kinetics (Plonka’s model), the effects of chain conformation on the decay kinetics of PVP• radicals); product analysis (classical methods, asymmetric flow field flow fractionation (AF4 or AFFFF); radiation-induced synthesis of select nanogels (radiation-induced synthesis of poly(acrylic acid) nanogels, polyvinylpyrrolidone nanogels, and binary polymer complexes - PAA and PVP); and possible medical applications of nanogels in the treatment of Alzheimer and tumor cells.

In the part Concluding Remarks the article's main findings and interpretations of analyzed data were summarized.

Authors cited 135 relevant references in the manuscript from which 24 of them are self-citation.

After careful reading, in my opinion, this manuscript needs some minor improvements according specific comments to be accepted for publication as follow:

  • Line 498: It is needed to check this sentence: "and the probability that more than a cell is occupied by. more than three polymer molecules is negligible."
  • Please, it is needed to provide full names of acronyms on first appearance in the main manuscript text and figure caption, e.g. line 558: "AFFFF", line 843: "IPC", line 874: "Bcl-2 siRNK", line 964: "PNIPAAm" and "LCST".
  • Line 675: It is needed to check this sentence: "Diffusion via. Brownian motion creates a counter-action motion to this force."
  • Lines 717-719: Please, it would be better to reconsider the sentence to be more simple for readers: "For the starting molecular weight of a few hundred thousand, and for the typical dose per pulse of ca. 1 kGy, a few pulses administered to a solution of 20 mM of repeating units yielded best results (only slight increase in Mw, pronounced decrease in Rg, no visible signs of degradation)."
  • Lines 963-971 should be moved in appropriate part of analysis in manuscript text, because, it is not usual to put a citation in the conclusion part.
  • Please, avoid 1st person plural  and rewrite all sentences to 3rd person plural, because it is common for scientific papers to be written in passive (lines: 101, 124, 161, 223, 288, 467, 473, 518, 669, 677).
  • Text font should be uniform, according Journal`s Instruction and template.
  • Please, it is needed to harmonize all cited literature according to Journal`s Instructions and template for authors, depending on the type of work in the References part.

After revision by the authors, this manuscript can be considered for publishing in the "Pharmaceutics" journal.

Best regards,

Reviewer

Author Response

We would like to thank the reviewer for their thorough review of our manuscript. We have addressed the raised points as follows:

Line 498: It is needed to check this sentence: "and the probability that more than a cell is occupied by. more than three polymer molecules is negligible."

New Sentence: ...and the probability that more than one cell is occupied by three or more polymer molecules is negligible. 

Please, it is needed to provide full names of acronyms on first appearance in the main manuscript text and figure caption, e.g. line 558: "AFFFF", line 843: "IPC", line 874: "Bcl-2 siRNK", line 964: "PNIPAAm" and "LCST".

Fixed: We provided the full names of the acronyms when they first appeared,
line 566: "...according to the Asymmetric Flow Field Flow Fractionation, AFFFF or AF4 measurements"

line 850-851: "...Ghaffarlou et al. published a comprehensive study on factors that affect the interpolymer complexes, IPC formation between PVP and PAA"

line 886: "...Doxorubicin or the RNA specific for the Bcl-2 gene (Bcl-2 siRNA)..."

line 976-977: "However, for polymers with low lower critical solution temperature (LCST)..."

line 978: replaced "PNIPAAm" with "poly(N-isopropylacrylamide)"

Line 675: It is needed to check this sentence: "Diffusion via. Brownian motion creates a counter-action motion to this force."

Changed to: line 683-684: "Diffusion through the Brownian motion of the particles creates a counter motion to the cross flow."

Lines 717-719: Please, it would be better to reconsider the sentence to be more simple for readers: "For the starting molecular weight of a few hundred thousand, and for the typical dose per pulse of ca. 1 kGy, a few pulses administered to a solution of 20 mM of repeating units yielded best results (only slight increase in Mw, pronounced decrease in Rg, no visible signs of degradation)."

Changed: We decided to take this sentence out as the example is not really necessary at this point.

Lines 963-971 should be moved in appropriate part of analysis in manuscript text, because, it is not usual to put a citation in the conclusion part.

We would like to leave this section in the conclusion.

Please, avoid 1st person plural  and rewrite all sentences to 3rd person plural, because it is common for scientific papers to be written in passive (lines: 101, 124, 161, 223, 288, 467, 473, 518, 669, 677).

Changed Lines 108 from "we will be discussing polymers in aqueous solutions." to "this review will focus on polymers in aqueous solutions."

Changed Lines 230 from "...(we may refer to these as nanogel particles) ." to "..(referred to as nanogel particles)"

Changed Lines 295 from "While in this work we focus on nanogel synthesis by radiation-induced... " to "While in this review will focus on nanogel synthesis by radiation-induced..."

Changed Lines 474-5 from "relates to inter-crosslinking or intra-crosslinking, we need to compare the measured decay time" to "...relates to inter-crosslinking or intra-crosslinking, the measured decay time needs to be compared with..."

Changed Lines 686-7 from "Putting these two forces together, we end up with smaller molecules moving faster" to "Putting these two forces together, the smaller molecules end up moving faster"

Text font should be uniform, according Journal`s Instruction and template.

We noticed some of the fonts were times new roman instead of palantino Linotype so we changed it so it is uniform.

We would like to thank the reviewer again for all their time and effort in reviewing this manuscript!

Reviewer 5 Report

This review paper deals with the mechanisms and kinetics of synthesizing polymer nanogels by ionizing-radiation-induced intramolecular crosslinking of macromolecules.

The structure of the manuscript is very good and the authors covered the scope of their work in depth.

I believe that this manuscript is suitable for publication. 

Author Response

We would like to thank the reviewer for their time and efforts in reviewing our manuscript!

Round 2

Reviewer 3 Report

The manuscript involves a review of ionizing-radiation-induced intramolecular crosslinking of macromolecules for the formation of nanogels. The topic is timely and the manuscript is well organized and written. The scope o the review is quite broad involving a description of ionizing radiation and of water radiolysis, the possible  radical reactions for the formation of nanogels, kinetics, nanogel characterization methods, radiation induced synthesis of PAA and PVP nanogels, as well as possible medical applications thereof. And the main points raised by the reviewers have been addressed. I recommend publication in the present form.